# Targeted Radionuclide Therapy Using Auger Electron Emitters: The Quest for the Right Vector and the Right Radionuclide

**DOI:** 10.3390/pharmaceutics13070980

**Published:** 2021-06-29

**Authors:** Malick Bio Idrissou, Alexandre Pichard, Bryan Tee, Tibor Kibedi, Sophie Poty, Jean-Pierre Pouget

**Affiliations:** 1Institut de Recherche en Cancérologie de Montpellier, Inserm U1194, Université de Montpellier, Institut Régional du Cancer de Montpellier, 34298 Montpellier, France; malick.bio-idrissou@inserm.fr (M.B.I.); alexandre.pichard@inserm.fr (A.P.); sophie.poty@inserm.fr (S.P.); 2Department of Nuclear Physics, Research School of Physics, Australian National University, Canberra, ACT 2601, Australia; Bryan.Tee@anu.edu.au (B.T.); Tibor.Kibedi@anu.edu.au (T.K.)

**Keywords:** Auger electrons, nuclear localisation sequence, NLS peptide, TAT peptide, radionuclide therapy

## Abstract

Auger electron emitters (AEEs) are attractive tools in targeted radionuclide therapy to specifically irradiate tumour cells while sparing healthy tissues. However, because of their short range, AEEs need to be brought close to sensitive targets, particularly nuclear DNA, and to a lower extent, cell membrane. Therefore, radioimmunoconjugates (RIC) have been developed for specific tumour cell targeting and transportation to the nucleus. Herein, we assessed, in A-431_CEA-luc_ and SK-OV-3_1B9_ cancer cells that express low and high levels of HER2 receptors, two ^111^In-RIC consisting of the anti-HER2 antibody trastuzumab conjugated to NLS or TAT peptides for nuclear delivery. We found that NLS and TAT peptides improved the nuclear uptake of ^111^In-trastuzumab conjugates, but this effect was limited and non-specific. Moreover, it did not result in a drastic decrease of clonogenic survival. Indium-111 also contributed to non-specific cytotoxicity in vitro due to conversion electrons (30% of the cell killing). Comparison with [^125^I]I-UdR showed that the energy released in the cell nucleus by increasing the RIC’s nuclear uptake or by choosing an AEE that releases more energy per decay should be 5 to 10 times higher to observe a significant therapeutic effect. Therefore, new Auger-based radiopharmaceuticals need to be developed.

## 1. Introduction

Targeted radionuclide therapy (TRT) is an attractive approach to treat cancer because it allows the specific irradiation of tumour cells. As, theoretically, TRT spares healthy tissues that do not express high levels of the targeted receptor, it represents a method of choice for treating diffuse and metastatic disease [1,2]. Historically, radiopharmaceuticals administered to treat patients have been based on beta-particle emitters, such as iodine-131 (^131^I) in thyroid carcinoma and phosphorus-32 (^32^P) in ovarian cancer. Later, [^90^Y]Y-ibritumomab tiuxetan (Zevalin) and [^131^I]I-tositumomab (Bexxar) were approved for radioimmunotherapy of non-Hodgkin B lymphoma. More recently, [^177^Lu]Lu-DOTATATE was also approved for therapy of neuroendocrine tumours [3], [^177^Lu]Lu-PSMA-617 for metastatic castration-resistant prostate cancer [4], and many clinical trials are currently assessing this radiopharmaceutical. Beta-particle emitters are largely available, at reasonable costs, and they are easy to chelate. However, although their relatively long range (≥1 mm) can counterbalance heterogeneity in radiopharmaceutical tumour distribution, it can also cause non-specific irradiation of healthy cells/tissues, thus limiting their interest for TRT. Moreover, beta particles are low linear energy transfer (LET) particles (like X and gamma rays). This means that they have a low ionising power and produce simple lesions in cells that can be repaired. Therefore, they can show limited efficacy against radioresistant solid tumours. This can be overcome by using high LET alpha particles, such as those emitted by radium-223 or by actinium-225, which have been tested for the management of prostate-cancer bone metastases. In addition, alpha-particle emitters decay according to a chain of disintegration, leading to several daughters that enhance therapeutic efficacy. Some questions remain, such as the choice of radionuclides, the stability of chelation, and the fate of daughters. Moreover, due to their range in matter (50–100 µm), alpha particles can also cause some non-specific irradiation. Auger electron emitters (AEE_S_), another class of therapeutic radionuclides, also have generated much interest and could mitigate the non-specific irradiation issue. Auger electrons are emitted in cascades by atoms decaying by electron capture and/or conversion processes that usually produce vacancies in K shells. These vacancies are filled by electrons dropping from the outer shells. These transitions are accompanied by a release of energy that can be transferred to electrons of external layers emitted in cascades (Auger electron cascades) or emitted as fluorescent X-rays.

The energy of Auger electrons can reach tens of keV, but most of them have a very low energy (<1 keV) that is released in a sphere of few cubic nanometres. Therefore, they are considered medium to high linear energy transfer (LET) particles (4–26 keV/µm). These high energy deposits produce locally dense ionisations that are very destructive when the radionuclide is incorporated in nuclear DNA, [5,6,7,8] the cell nucleus [9,10,11,12,13], or the cell membrane [14,15,16,17,18,19]. In a context where specific nuclear DNA targeting is still challenging, our group previously showed that AEE-based TRT with ^125^I-labelled non-internalising monoclonal antibodies (mAbs) against carcinoembryonic antigen (CEA) leads to significant tumour growth delay in preclinical mouse models harbouring small tumours [18]. While nuclear targeting directly mediates the DNA damage response (DDR) leading to cell death, Auger electron cascades produced at the cell membrane induce the local formation of ceramide-enriched large domains (lipid rafts) that activate signalling pathways involving NF-kB and ultimately, leading to reactive oxygen species (ROS) production, nuclear damage, and DDR activation [15,19]. We also demonstrated that bystander effects induced by nuclear (5-[^125^I]iodo-2′-deoxyuridine, [^125^I]I-UdR) and cell membrane (non-internalising ^125^I-mAbs) irradiation contribute significantly to cell death [13,15]. Notably, we showed that, after incubation with the DNA base analogue [^125^I]I-UdR, 30–35% of cells were killed by bystander effects and 50% of cells by the direct effect of Auger electrons.

AEEs could be a precious asset for TRT, but there has not been, so far, any convincing candidate, although about 50% of radionuclides emit AEs. Indeed, a major advantage of AEE-based TRT is to decrease the non-specific irradiation of healthy tissues, while benefiting from Auger electrons’ high LET features. However, due to their short range and lack of cross-fire irradiation in tumours, AAE-based TRT might require the administration of high activities or repeated administrations to counterbalance the radionuclide distribution heterogeneity that would not allow reaching all tumour cells after one single administration. Therefore, ideal AEE should not emit (or may emit only in very low proportion) other radiation types (γ, β, CE), responsible for non-specific irradiations, as these would overwhelm AEE-TRT’s benefits. AEE also need to locally release high amount of energy, a parameter that is influenced by the number of Auger electrons released per decay and their average energy. Iodine-125, which emits about 20 Auger electrons/decay and releases low amount of energy through the emission of conversion electrons (CE) or photons, is a very attractive AEE candidate. Iodine-125 radiopharmaceuticals have been developed ([^125^I]I-UdR or mAbs conjugates) [20,21,22,23], but their clinical use is limited by ^125^I’s long physical half-life (t_1/2(Phys)_ = 59.4 days). A comprehensive list of AEE candidates has recently been reviewed in [24]. Platinum-193 m (t_1/2(phys)_ = 4.3 days) and platinum-195 m (t_1/2(phys)_ = 4.0 days) also are very attractive AEEs because they emit a very high number of Auger electrons (between 20 and 30) per decay and photons of intermediate energy (66 and 76 keV) than can be used for imaging. However, they are still not routinely available, and the radiolabelling of biological vectors is still challenging. Many studies have used indium-111 because it has a short physical half-life (2.8 days), is easily conjugated to macromolecules, and is easily produced [25]. However, ^111^In is a poor AEE with about 7–8 Auger electrons per decay, emission of energetic CEs (0.16 per decay; average energy = 176 keV) and γ-rays (1.85 energetic photons of 171.28 and 245.35 keV per decay) that can cause non-specific irradiation and raise radiation protection concerns, respectively.

Another difficulty with AEEs is the need to build radioimmunoconjugates (RIC) that can specifically target cancer cells and also drive activity at least into the nucleus, if not into DNA. Indeed, as Auger electrons with high LET have a range of few nm, nuclear localisation without incorporation into DNA would lead to a significant decrease in efficacy, although the results might still be acceptable.

Herein, we assessed two new ^111^In-RICs based on the anti-HER2 antibody trastuzumab functionalised with two cationic peptides (NLS and TAT) harbouring a nuclear localising sequence. Pioneering groups have shown that these peptides allow the nuclear transportation of the functionalised antibodies in AEE-based TRT settings [11,26,27]. Although, other authors highlighted the need for alternative approaches to improve their delivery [28,29,30], none of these approaches has been evaluated beyond the preclinical stage. Moreover, only one study investigated [^1^^11^In]In-DTPA-human epidermal growth factor in patients [31] but did not confirm the promising results obtained in animal models [28]. In this study, we evaluated whether NLS-/TAT-immunoconjugates are suitable vectors to specifically deliver AEE-based TRT to the cancer cell nucleus. We hypothesised that the ^111^In-trastuzumab RIC specifically targets HER2-positive cancer cells before internalization in the cell cytoplasm. When conjugated to NLS or TAT peptides, the nuclear localisation signal should drive this RIC through the nuclear pore complex to enhance Auger electron cytotoxicity. To test this hypothesis, we thoroughly investigated the subcellular localisation, activity uptake, and cytotoxic effects of the developed ^111^In-RICs.

## 2. Materials and Methods

### 2.1. Cell Lines

A-431 vulvar squamous carcinoma and SK-OV-3 ovarian carcinoma cells were obtained from the American Type Culture Collection (ATCC, Rockville, MD, USA). HER1- and HER2-expressing A-431 cells were transfected with the genes encoding CEA and luciferase to obtain the A-431_CEA-luc_ cell line, as previously described [14]. HER1- and HER2-expressing SK-OV-3 cells were transfected with the gene encoding CEA to obtain the SK-OV-3_1B9_ cell line. All cell lines were grown in DMEM supplemented with 10% heat-inactivated foetal bovine serum, 1% penicillin/streptomycin, and 200 µg/mL geneticin. Hygromycin (100 µg/mL) was added to the medium for A-431_CEA-luc_ cells. Cell lines were routinely tested for mycoplasma contamination using the MycoTect assay from Life technologies (Thermo Fisher Scientific, Waltham, MA, USA).

### 2.2. Antibodies and Peptides

Trastuzumab (*Herceptin*, Roche, Basel, Switzerland) is a humanised IgG1k mAb against the HER2 receptor. A humanised anti-AMHRII IgG1 mAb, 3C23K, was used as an irrelevant mAb, derived from the murine anti-AMHRII mAb 12G4 [32].

The 16-mer TAT peptide [GRKKRRQRRRPPQGYG] harbouring the nuclear localisation sequence (underlined) of the HIV-1 tat protein was synthesised and provided by Genscript Biotech (Leiden, The Netherlands) with a purity ≥ 90%. The 13-mer Simian Virus 40 T-Ag-derived NLS peptide [CGYGPKKKRKVGG] (the nuclear localisation sequence is underlined) was synthesised and provided by Proteogenix (Schiltigheim, France) with a purity of 90.40%. Both peptides were redissolved in chelexed PBS (pH 7.2) at a concentration of 7 mg/mL and kept at −80 °C until use.

### 2.3. Bioconjugation, Radiolabelling, Purification and Quality Control

Trastuzumab and the irrelevant humanised IgG (3C23K) were conjugated to *p*-*SCN*-*Bn*-*CHX*-A″-*DTPA (*[(R)-2-amino-3-(4-isothiocyanatophenyl)propyl]-trans-(S,S)-cyclohexane-1,2-diamine-pentaacetic acid), provided by Macrocyclics (Plano, TX, USA), a chelating agent allowing further radiolabelling with ^111^In. Briefly, the pH of the trastuzumab and IgG solutions (2–8 mg in chelexed PBS; 8–12 mg/mL) was adjusted to 8.4 using a 0.2 M chelexed Na_2_CO_3_ solution (pH 10). A 15-fold molar excess of *p*-*SCN*-*Bn*-*CHX*-A″-*DTPA* (25 mg/mL in DMSO) was added drop by drop (2 µL) to prevent precipitation. Bioconjugation was performed at 37 °C for 90 min, followed by purification on an Amicon 30 kDa (Merck Millipore, Molsheim, France) and chelexed PBS washes (pH 7.2).

The NLS and TAT peptides were conjugated to DTPA-modified trastuzumab or IgG using the heterobifunctional cross-linking agent sulphosuccinimidyl-4-(N-maleimidomethyl)cyclohexane-1-carboxylate (sulpho-SMCC; Thermo Fisher Scientific, Rockford, IL, USA). Briefly, DTPA-trastuzumab or DTPA-IgG (chelexed PBS, pH 7.2) was mixed with 25- to 50-fold molar excess of sulpho-SMCC dissolved to 2.34 mg/mL in Ultra Trace Elemental Analysis Grade water (Thermo Fisher Scientific, Illkirch-Graffenstaden, France) at room temperature (RT) for 60 min. The obtained activated-ester DTPA-trastuzumab or IgG conjugates were transferred to Amicon 30 kDa (Merck Millipore, Burlington, MA, USA), purified, and concentrated to 3–7 mg/mL with chelexed PBS (pH 7.5). Then, conjugates were mixed with 40- to 60-fold molar excess of NLS or TAT peptides (7 mg/mL in chelexed PBS [pH 7.2]) at 37 °C for 90 min. The final conjugates were transferred to Amicon 30 kDa (Merck Millipore), and the PBS buffer was exchanged to ammonium acetate (0.25 M, pH 5.5) with two consecutive washes.

Matrix-assisted laser desorption ionisation–time of flight (MALDI–TOF) was used to determine the number of DTPA, SMCC, NLS, or TAT moieties per antibody. The m/z difference between the native mAbs and their conjugates was divided by the molecular weight of the added functional group (DTPA, sulpho-SMCC, or peptide) to find the chelate/peptide-to-antibody ratio.

HER2-specific immunoconjugates (DTPA-trastuzumab, DTPA-trastuzumab-NLS, and DTPA-trastuzumab-TAT) (Figure 1a) and irrelevant immunoconjugates (DTPA-IgG, DTPA-IgG-NLS, and DTPA-IgG-TAT) were labelled with indium-111 ([^111^In]InCl_3_) to obtain a specific activity of 370 MBq/mg. Typically, 119 µL (100 MBq) of [^111^In]InCl_3_ (in HCl 0.05 N; Curium, Paris, France) were mixed with 119 µL 0.25 M ammonium acetate (pH 5.5) to stabilise the pH at 5.5. Then, 270 µg (volume (µL), depending on the concentration) of immunoconjugate were added at 37 °C for 90 min. Radiolabelling yields and radiochemical purity were determined by instant thin-layer silica-gel chromatography (iTLC, Montpellier, France) with 50 mM EDTA (pH 5.5) as eluant. RICs were purified using a desalting PD-10 column and eluted with PBS. All RICs were successfully obtained, with radiochemical yields between 80 and 98% and specific activities ranging from 310 to 380 MBq/mg.

### 2.4. Immunoconjugates Cell Binding and HER2 Cell Expression

Immunoconjugate cell binding was assessed by flow cytometry. Cells were fixed with 4% (*v/v*) paraformaldehyde in PBS at RT for 15 min, permeabilised with Triton X-100 (0.5% in PBS) at RT for 15 min, and blocked with PBS/bovine serum albumin (BSA) blocking solution (10 mg/mL) at 37 °C for 1 h. Then, cells were incubated with each immunoconjugate (10 µg/mL) at RT for 1 h, followed by incubation with an AlexaFluor-488-conjugated anti-human secondary antibody (1:200; (F(ab′)_2_-goat anti-human IgG (H+L) Cross-Adsorbed, Thermo Fisher Scientific) at RT in the dark for 1 h. Cells were then washed three times and suspended in PBS for analysis using a Gallios cytometer (Beckman Coulter, Villepinte, France).

### 2.5. RIC Stability

A 100 µL aliquot of each radiopharmaceutical was added to 900 µL of culture medium or human serum. Then, samples were incubated on a shaker at 37 °C. At the desired time points, the radiopharmaceutical stability was evaluated by iTLC using a 50 mM EDTA (pH 5.5) solution as eluant. The retention factor (Rf) of the RIC was 0 and that of free indium was 1. Each experiment was repeated twice and the percentage of radionuclides bound to proteins for each RIC was calculated after measurement with a γ-ray counter.

### 2.6. Determination of the RIC Immunoreactive Fraction

RIC immunoreactive fractions were assessed using a bead-based assay method described by Sharma et al. [33]. Briefly, 20 µL of the magnetic HisPur Ni-NTA bead slurry (Thermo Scientific) was incubated with 1 µg His-Tag human recombinant HER2 protein (Sino Biological). Coated beads were then incubated with 1 ng of RIC in the presence (or not) of unlabelled trastuzumab (blocked condition; ×1000-fold excess) on a shaker at RT for 30 min. Beads were isolated, and the supernatant containing unbound RICs was collected. Next, beads were washed twice with PBS-Tween (0.05%) to remove non-specifically bound RICs. Finally, the activity in each fraction (beads alone, supernatant, and wash) was measured using a γ-ray counter, and the immunoreactive fractions were calculated.

### 2.7. Clonogenic Assays

Cell survival after incubation with ^111^In-RICs or unlabelled immunoconjugates was assessed using a standard clonogenic assay. Briefly, a defined number of cells, from 100 to 800, were plated in 6-well plates in 2 mL of fresh culture medium. The next day, cells were incubated with an increasing amount of activities of ^111^In-RICs (0 to 4 MBq/mL) or 10.8 µg/mL unlabelled immunoconjugates at 37 °C/5% CO_2_ for 48 h, followed by three washes with 2 mL PBS and the addition of fresh medium (2 mL). Then, cells were cultured for 12 days, and colonies were fixed (acetic acid:methanol [1:3]) and stained with Giemsa diluted in H_2_O (1:5; Sigma-Aldrich, Saint-Quentin Fallavier, France). Colonies were manually counted, and the surviving fraction was calculated. All experiments were repeated at least three times in triplicate for each condition.

### 2.8. In Vitro Measurement of DNA Damage

Cells were seeded on coverslips in 6-well plates. The next day, cells were incubated with 4 MBq/mL of HER2-specific or irrelevant ^111^In-RICs for 48 h. After RIC removal, cells were washed with PBS three times and fixed with 4% (*v/v*) paraformaldehyde in PBS at RT for 15 min, permeabilised with Triton X-100 (0.5% in PBS) at RT for 15 min, and blocked with PBS/BSA blocking solution (10 mg/mL) at 37 °C for 1 h. Next, cells were incubated with an anti-phosphorylated histone H2AX (γH2AX) antibody diluted in PBS (1:200; 05-636, Sigma Aldrich, Saint-Quentin Fallavier, France) at 37 °C for 1 h, followed by washing and incubation with a goat anti-mouse IgG (H+L) FITC-conjugated antibody diluted in PBS/BSA (1:400; 12-506, Sigma-Aldrich, Saint-Quentin Fallavier, France) at 37 °C for 1 h. Finally, cells were washed with PBS/Tween-20 (0.1%) and then with PBS (three times/each). Coverslips were mounted on slides with Vectashield antifade mounting medium and DAPI (Vector laboratories, Les Ulis, France) and imaged using a ×63 or ×40 objective on a Leica fluorescence inverted microscope (Leica Microsystems, Wetzlar, Germany), as described previously [15].

### 2.9. Radioactivity Uptake and Cell Fractionation

To isolate nuclei, the chaotropic method described by Pouget et al. [34] was used. Briefly, 1 × 10^6^ cells grown in T25 flasks overnight were incubated with 0.1 MBq/mL (3 mL) ^111^In-RICs for 48 h. At various time-points (1, 2, 3, and 6 days), cells were trypsinised and rinsed with PBS three times to remove unbound RICs. The activity of isolated whole cells was determined with a γ-ray counter (Hidex Automatic gamma counter, Villebon-sur-Yvette, France). Next, cells were resuspended in 1 mL lysis buffer (320 mM sucrose, 5 mM MgCl_2_, 10 mM Tris-HCl, 0.1 mM deferoxamine, and 1% Triton X-100 in H_2_O) and centrifuged at 1500× *g* for 10 min. The supernatant (cytoplasm + membrane) was collected. The pellet was again resuspended in 1 mL lysis buffer, centrifuged at 1500× *g* for 10 min, and the supernatant was collected. The pellet (cell nuclei) was rinsed with 1 mL PBS three times to remove all traces of cytoplasm or membrane. Finally, the activity present in both fractions (nuclei and cytoplasm + membrane) was measured using a γ-ray counter. The cumulative number of decays (Bq.s) per cell over 6 days was then calculated as well as the cumulative number of decays (Bq.s) per fraction (nucleus and cytoplasm + membrane).

### 2.10. Evaluation of Nuclear Fraction Contamination by Western Blotting

Nuclei were rinsed twice with PBS and lysed in RIPA buffer (Santa Cruz; Santa Cruz, CA, USA) at 4 °C for 30 min. The total protein concentration in the nuclear fraction was determined using the BCA Protein Assay Reagent (BCA Assay Kit, Thermo Fisher scientific, Illkirch, France). Then, 6 µg of nuclear proteins were separated by SDS-PAGE (12% poly-acrylamide gel) and electrotransferred onto nitrocellulose membranes (Bio-Rad, Grabels, France). Membranes were incubated with anti-α-tubulin (11H10) (1/1000; Cell Signaling Technology, Leiden, The Netherlands) and anti-phosphorylated histone 3 (Ser10) antibodies (1/1000; Cell Signaling Technology, Leiden, The Netherlands) diluted in Tris-buffered saline/BSA (5%). A horseradish peroxidase-conjugated anti-rabbit IgG (Cell Signaling Technology, Danvers, MA, USA) was used as a secondary antibody. Proteins were revealed using an enhanced chemiluminescence system according to the manufacturer’s instructions (Clarity, Bio-Rad, Marnes-La-Coquette, France). Protein expression was imaged with a PXi analyser (Ozyme, St Quentin en Yvelines, France).

### 2.11. Immunofluorescence Analysis of the Immunoconjugate Subcellular Localisation

As described in Section 2.8, cells were plated on coverslips in 6-well plates, and, the day after, they were incubated with different unlabelled immunoconjugates for 48 h. Then, cells were fixed and permeabilised (see Section 2.8). After blocking with PBS/BSA, an AlexaFluor-488-conjugated anti-human secondary antibody (1:200; (F(ab′)_2_-goat anti-human IgG (H+L) Cross-Adsorbed, Thermo Fisher Scientific) was added at 37 °C in the dark for 1 h. Coverslips were then mounted with Vectashield with DAPI (see Section 2.8) and imaged using a ×63 or ×40 objective and a Leica fluorescence inverted microscope.

### 2.12. Energy Deposition Calculations

The energy deposition of Auger electrons, CEs, gamma- and X-rays in spheres was evaluated using the PENELOPE code system [35] that simulates the interactions of radiations as they pass through the medium, in this case water. The detailed energy spectra of the radiations were generated with the BrIccEmis [36,37] code and typically using 5 million decay events. All simulations were done using the Monte Carlo approach.

### 2.13. Statistical Analysis

Data were analysed using GraphPad Prism version 8.4.3 (686). Data were described using means and standard deviation (SD). For gamma-H2AX, data were described using means and the standard error of the mean (SEM). Data were compared using the Student’s *t* test. A *p* value < 0.05 was considered significant. The Bliss independence model was used to determine the percentage of specific irradiation. In this method, the global efficacy of [^111^In]In-trastuzumab was dissociated in specific and non-specific efficacy using the following formula, wherein the non-specific efficacy was determined as [^111^In]In-IgG efficacy:$$Efficacy_{\left[111\mathrm{In}\right]\mathrm{In}-\mathrm{trastuzumab}\;}=Specific\;efficacy+Non\;specific\;efficacy-\left(Specific\;efficacy\times Non\;specific\;efficacy\right)$$

## 3. Results

### 3.1. Characterisation of Immunoconjugates and Radiolabelling

After conjugation with a 15-fold molar excess of p-SCN-Bn-CHX-A″-DTPA, 2 ± 1 DTPA per mAb were obtained, on average, as confirmed by MALDI–TOF analysis (Figure 1b,c). Subsequent bioconjugation with 50-fold molar excess of the reticulating agent sulpho-SMCC and 60-fold molar excess of NLS peptides resulted in a mean of 17–22 NLS peptides per mAb. These immunoconjugates will be referred to as trastuzumab-NLS_17-22_. This number decreased to 5–10 NLS peptides per mAb when a 25-fold molar excess of sulpho-SMCC or a 40-fold molar excess of NLS peptides were used. These immunoconjugates will be referred to as trastuzumab-NLS_5-10_. Concerning the functionalisation with TAT peptides, 2 ± 1 TAT peptides/mAb were obtained after conjugation with 50-fold molar excess of sulpho-SMCC followed by 60-fold molar excess of TAT peptides. These immunoconjugates will be referred to as trastuzumab-TAT_1-3_.

The in vitro stability of RICs was assessed by incubation in cell-culture medium and human serum at 37 °C. Results showed excellent stability with percentages of ^111^In bound to protein > 82% over 72 h (Figure 2a).

Flow cytometry analysis confirmed that HER-2 expression was higher in SK-OV-3_1B9_ cells (G-mean: ×116 compared to control) than in A-431_CEA-luc_ cells (G-mean: ×4.93 compared to control) (Figure 2b). Immunoreactivity assay confirmed that the binding of NLS- or TAT-immunoconjugates was predominantly non-specific because [^111^In]In-IgG-NLS_5-10_ and [^111^In]In-IgG-TAT_1-3_ bound to HER2-coated beads. Moreover, the binding of [^111^In]In-trastuzumab-NLS_5-10_ and [^111^In]In-trastuzumab-TAT_1-3_ was not strongly blocked in the presence of a large excess of unlabelled trastuzumab compared with [^111^In]In-trastuzumab (Figure 2c, left panel). Flow cytometry analysis of the immunoconjugate cell binding also showed that trastuzumab functionalisation with NLS_5-10_ or TAT_1-3_ peptides resulted in an increase of the fluorescence signal (G-mean: 104,042, 506,000, and 265,795 for trastuzumab, trastuzumab-NLS_5-10_, and trastuzumab-TAT_1-3_, respectively), indicating higher cell binding. However, this effect was predominantly non-specific because it was observed also after functionalisation of the non-specific IgG mAb (Figure 2c, right panel).

### 3.2. Functionalisation with TAT_1-3_ Peptides Is Associated with a Moderate, Non-Specific Cytotoxicity Increase

Incubation of A-431_CEA-luc_ and SK-OV-3_1B9_ cells with 10.8 µg unlabelled mAb, mAb-NLS_5-10_, or mAb-TAT_1-3_ (corresponding to the mAb mass added in the ^111^In-treated 4 MBq/mL groups) for 48 h did not have any effect on clonogenic survival compared with untreated cells (Appendix A). Conversely, the clonogenic survival of cells exposed to increasing activities (0–4 MBq/mL) of [^111^In]In-trastuzumab or [^111^In]In-IgG was significantly decreased (Figure 3). At 4 MBq/mL, survival was 54 ± 4% (*p* < 0.0001) and 64 ± 4% (*p* < 0.0001) after incubation with [^111^In]In-trastuzumab, and 71 *± 3*% (*p* < 0.0001) and 75 ± 2% (*p* < 0.0001) after incubation with [^111^In]In-IgG, in A-431_CEA-luc_ and SK-OV-3_1B9_ cells, respectively. This indicates that non-specific ^111^In irradiation contributes to 25–29% of cell death, while specific irradiation kills between 15 and 24% more cells (using the Bliss independence model).

Energy deposit calculations indicated that ^111^In emitted 7.19 Auger electrons per decay with a mean energy of 0.94 keV released over a maximal range of about 12 µm. It also emitted 0.16 CEs per decay, which corresponded to a mean energy of 176 keV (maximum range of 573 µm). The energy deposition in a sphere of 20 µm, which corresponds roughly to the diameter of a SK-OV-3_1B9_ cell, was about 5.97 keV. This increased up to 37 keV for a sphere of 100 µm in diameter. ^111^In also emitted 1.85 energetic photons of 150.8 and 245.3 keV per decay, depositing only about 0.114 keV in a 100 µm diameter sphere that did not contribute to cell killing in vitro. Therefore non-specific cytotoxicity could be due to CEs co-emitted by ^111^In.

Functionalisation with TAT peptides ([^111^In]In-trastuzumab-TAT_1-3_; Figure 1a) did not improve cytotoxicity in A-431_CEA-luc_ cells (*p* = 0.4; Figure 3a). In SK-OV-3_1B9_ cells, it increased cytotoxicity only at the highest tested volumic activity (4 MBq/mL) compared with [^111^In]In-trastuzumab (44 ± 3% vs. 64 ± 4%, *p* = 0.005; Figure 3a). At this volumic activity, [^111^In]In-IgG-TAT_1-3_ showed similar cytotoxicity (29 ± 0.4%, *p* = 0.1). A similar trend for [^111^In]In-IgG-TAT_1-3_ was observed at all tested volumic activities, indicating that the functionalisation of the trastuzumab-DTPA conjugate with TAT peptides is associated with loss of antigen specificity.

### 3.3. Functionalisation with NLS_17-22_ Peptides Is Associated with Higher Non-Specific Cytotoxicity

Then, trastuzumab-DTPA was conjugated to NLS peptides (Figure 1a), and the cytotoxicity of the resulting [^111^In]In-trastuzumab-NLS_17-22_ was assessed. Exposure to [^111^In]In-trastuzumab-NLS_17-22_ resulted in a marked decrease in cell survival at all tested volumic activities, but, again, a similar trend was observed when using the non-specific [^111^In]In-IgG-NLS_17-22_ (Figure 3b)_._

### 3.4. Functionalisation with NLS_5-10_ Is Associated with High and Moderate Non-Specific Cytotoxicity

On the basis of the hypothesis that the non-specific cytotoxicity of [^111^In]In-trastuzumab-NLS_17-22_ could be due to the mAb’s high degree of functionalisation with NLS peptides, [^111^In]In-trastuzumab-NLS_5-10_ (only 5 to 10 NLS peptides per mAb molecule) was produced (Figure 1a). At 4 MBq/mL, [^111^In]In-trastuzumab-NLS_5-10_ significantly decreased cell survival to 18 ± 3% (*p* = 0.001) and 23 ± 0.6% (*p* = 0.0003), compared with [^111^In]In-trastuzumab, in A-431_CEA-luc_ and SK-OV-3_1B9_ cells, respectively (Figure 3c). [^111^In]In-IgG-NLS_5-10_ cytotoxic activity was significantly lower than that of [^111^In]In-trastuzumab-NLS_5-10_ in both cell lines (*p* = 0.01 for A-431_CEA-luc_ and *p* = 0.02 for SK-OV-3_1B9_) and also compared with [^111^In]In-IgG-NLS_17-22_. However, [^111^In]In-IgG-NLS_5-10_ cytotoxic activity remained comparable to that of [^111^In]In-trastuzumab (*p* = 0.1) in A-431_CEA-luc_.

### 3.5. [^111^In]In-Trastuzumab-NLS_5-10_ Is Associated with the Highest Activity Uptake per Cell or Nucleus

A nucleus-isolation technique was used to determine the activity incorporated into the cell nucleus and the activity remaining in the extranuclear fraction (cell cytoplasm and membrane). As these experiments required high cell numbers and large volumes, the activity of 0.1 MBq/mL was chosen. Western blot analysis showed that α-tubulin (marker of the cytoplasmic fraction) was not or barely detectable, whereas histone H3 (nuclear fraction marker) was strongly expressed in the nuclear fraction of both cell lines (Figure 4a), validating our nucleus-isolation technique (<4% contamination from the cytoplasmic fraction).

Then, the nuclear and extranuclear activity curves showed a similar trend in both cell lines, with a progressive increase and maximal uptake values at 24 or 48 h after RIC addition (Figure 4b). As HER2 receptor expression is higher in SK-OV-3_1B9_ cells (Figure 2b), uptake was approximately one log higher in this cell line with a slower activity decrease at 72 h (i.e., 24 h after radioactivity removal) (Figure 4b).

From these curves, the cumulative number of decays (Ã) occurring in the whole cell (Ã_cellular_), in the cytoplasm + membrane (Ã_extranuc_), and in the nucleus (Ã_nuclear_) over 6 days was determined (Figure 4c). In the absence of a specific peptide sequence targeting the cell nucleus, Ã_nuclear_ and Ã_cellular_ for [^111^In]In-trastuzumab were 128 and 898 Bq.s in A-431_CEA-luc_ cells and 10 times higher or more (1503 and 12,704 Bq.s) in SK-OV-3_1B9_ cells. The Ã_cellular_ values for [^111^In]In-IgG were <59 Bq.s in A-431_CEA-luc_ and <410 Bq.s in SK-OV-3_1B9_ cells. The addition of NLS_5-10_ or TAT_1-3_ sequences globally improved RIC subcellular uptake (in all compartments) and in both cell lines with a more pronounced effect from NLS_5-10_ than from TAT_1-3_. HER2 expression level (high versus low) influenced the Ã_cellular_ values, and also the Ã_nuclear_ values, which were multiplied by 6.1 in A-431_CEA-luc_ (low expression; trastuzumab versus trastuzumab-NLS_5-10_) and only by 2.0 in SK-OV-3_1B9_ (high expression; trastuzumab versus trastuzumab-NLS_5-10_).

### 3.6. TAT and NLS Peptides Do Not Increase the Percentage of Activity Reaching the Nucleus

Although, the raw Ã_extranuc_ and Ã_nuclear_ values increased with the addition of NLS and TAT peptides, the ratio between these values did not highlight any specific increase in nuclear targeting (Figure 5a).

The proportion of radioactivity in the nucleus remained between 6 and 18% for all RICs, indicating that nuclear targeting was not specific. It must be noted that the nuclear proportion of [^111^In]In-IgG-TAT_1-3_ was higher (37% and 40% in A-431_CEA-luc_ and SK-OV-3_1B9_ cells, respectively), but this was mainly due to low cellular uptake such that the percentage was increased. The same conclusion can be drawn for [^111^In]In-IgG in A-431_CEA-luc_ (Figure 5a).

By analogy, for [^111^In]In-trastuzumab-NLS_5-10_, the Ã_extranuc_ value represented 81% (versus 86% for [^111^In]In-trastuzumab) of the cumulative number of decays (Ã) occurring in the whole cell (Ã_cellular_) in A-431_CEA-luc_ and 87% (versus 88% for [^111^In]In-trastuzumab) in SK-OV-3_1B9_ cells, indicating that NLS addition improved cellular uptake but not nuclear targeting (Figure 5b).

For [^111^In]In-trastuzumab-TAT_1-3_, Ã_extranuc_ represented 86% in A-431_CEA-luc_ (versus 86% for [^111^In]In-trastuzumab) and 85% in SK-OV-3_1B9_ (versus 88% for [^111^In]In-trastuzumab) of all decays. Therefore, as observed for NLS_5-10_, the addition of TAT_1-3_ did not significantly improve nuclear targeting but did increase the total cellular uptake.

### 3.7. TAT and NLS Increase the Non-Specific Cellular Uptake

The addition of NLS_5-10_ or TAT_1-3_ resulted in an overall increase of cellular, and subsequently nuclear, uptake of the non-specific IgG RIC (Figure 4c and Figure 5). Compared with [^111^In]In-IgG, the Ã_cellular_ value for [^111^In]In-IgG-NLS_5-10_ was 52 times higher in A-431_CEA-luc_ cells (versus 4.7 times for [^111^In]In-trastuzumab-NLS_5-10_ compared with [^111^In]In-trastuzumab) and 37 times in SK-OV-3_1B9_ cells (versus 1.8 times for [^111^In]In-trastuzumab-NLS_5-10_, compared with [^111^In]In-trastuzumab). For [^111^In]In-IgG-TAT_1-3_, the Ã_cellular_ value was 26 times higher in A-431_CEA-luc_ cells (versus 3.2 times for [^111^In]In-trastuzumab-TAT_1-3_, compared with [^111^In]In-trastuzumab) and 9.2 times higher in SK-OV-3_1B9_ cells (versus 1.1 times for [^111^In]In-trastuzumab-TAT_1-3_, compared with [^111^In]In-trastuzumab). The HER2-specific/non-specific Ã_cellular_ ratios for NLS_5-10_ were 1.4 in A-431_CEA-luc_ and 1.5 in SK-OV-3_1B9_ cells. The HER2-specific/non-specific Ã_cellular_ ratios for TAT_1-3_ were 1.9 in A-431_CEA-luc_ and 3.7 in SK-OV-3_1B9_ cells.

### 3.8. Immunoconjugates Do Not Localise in the Nucleus

Although less sensitive than radioactive detection, the localisation of the immunoconjugates was next assessed by immunofluorescence analysis (Figure 6). DTPA-trastuzumab was detected at the cell surface and in the cytoplasm. The amount of foci in the cytoplasm increased for DTPA-trastuzumab-NLS_5-10_ and DTPA-trastuzumab-TAT_1-3_. Conversely, DTPA-IgG mAb was not detected at the cell surface or in the cytoplasm. However, when functionalised with NLS_5-10_ or TAT_1-3_, the number of cytosolic DTPA-IgG mAb foci increased. Few foci of NLS- and TAT-functionalised mAbs were detected in the nucleus, but their number was much lower than in the cytoplasm.

### 3.9. [^111^In]In-Trastuzumab-NLS_5-10_ Induces the Highest Number of γH2AX Foci per Nucleus

To investigate whether DNA double strand breaks (DSBs) induction by the different RICs was correlated with their Ã_cellular_ or Ã_nuclear_ values, γH2AX focus formation was monitored at various time points in A-431_CEA-luc_ and SK-OV-3_1B9_ cells by immunofluorescence analysis (Figure 7a,b, left panels). The highest yield of γH2AX foci (indicative of DNA DSBs) was observed in cells incubated with [^111^In]In-trastuzumab-NLS_5-10_ (12.0 *±* 7.0 foci/cell and 9.9 ± 2.9 foci/cell in A-431_CEA-luc_ and SK-OV-3_1B9_ cells, respectively). In A-431_CEA-luc_ cells, it was followed by [^111^In]In-IgG-NLS_5-10_ (5.0 *±* 4.1 foci/cell), [^111^In]In-trastuzumab (4.1 *± 5.9* foci/cell, and [^111^In]In-trastuzumab-TAT_1-3_ (3.5 *± 5.9* foci/cell). In SK-OV-3_1B9_ cells, it was followed by [^111^In]In-trastuzumab-TAT_1-3_ (7.7 *±* 3.4 foci/cell), [^111^In]In -IgG-TAT_1-3_ (5.9 *±* 3.5 foci/cell), [^111^In]In-IgG-NLS_5-10_ (5.3 *±* 2.4 foci/cell), and [^111^In]In-trastuzumab (3.3 *±* 2.3 foci/cell). For comparison, 0.4 ± 0.7 foci/cell and 2.0 ± 2.1 foci/cell were detected in untreated cells and in A-431_CEA-luc_ cells incubated with [^111^In]In-IgG, respectively. The corresponding values were 0.6 ± 0.6 foci/cell and 2.6 *±* 2.4 foci/cell in SK-OV-3_1B9_ cells. Then, the cumulative number of γH2AX foci over 144 h was calculated and expressed as a function of the Ã_cellular_ or Ã_nuclear_ values (Figure 7a,b, right panels). Such foci increased with time, and the *r*^2^ values ranged between 0.43 and 0.75. Neither Ã_cellular_ nor Ã_nuclear_ appeared to be the best parameter.

## 4. Discussion

This study assessed the ability of two highly cationic peptides, the synthetic 13-mer NLS peptide [CGYGPKKKRKVGG] from the simian virus 40 large-T antigen and the synthetic 16-mer TAT peptide [GRKKRRQRRRPPQGYG] from the HIV-1 TAT protein, to drive the ^111^In-anti-HER2 (trastuzumab) RIC into the cell nucleus of two HER2-expressing cancer cell lines, A-431_CEA-luc_ (HER2^+^) and SK-OV-3_1B9_ (HER2^++^). According to the literature, after HER2-mediated internalisation in the cytoplasm, NLS and TAT peptide sequences are recognised by importin-α and importin-β1 to form a nuclear-pore-targeting complex [38,39]. This complex facilitates RIC passage through the nuclear pore and their translocation into the nucleus where they will be dissociated to release NLS-/TAT-conjugates [11]. In total, eight RICs were obtained and characterised: [^111^In]In-trastuzumab, [^111^In]In-trastuzumab-NLS_5-10_, [^111^In]In-trastuzumab-NLS_17-22_, [^111^In]In-trastuzumab-TAT_1-3_ as a specific anti-HER2 RIC, and [^111^In]In-IgG, [^111^In]In-IgG-NLS_5-10_, [^111^In]In-IgG-NLS_17-22_, and [^111^In]In-IgG-TAT_1-3_ as non-specific RICs.

First, functionalisation of trastuzumab with the TAT or NLS peptides slightly increased its cell binding, but this binding was at least predominantly mediated by non-specific mechanisms. Moreover, non-radioactive immunoconjugates showed very low cytotoxicity, even when used at the highest concentration (10.8 µg/mL; equivalent to tested volumic activity of 4 MBq/mL for ^111^In-RIC) (Appendix A).

Then, comparison of the different RICs showed that [^111^In]In-IgG was significantly less cytotoxic than [^111^In]In-trastuzumab, but more than what was reported in previous studies. The origin of ^111^In non-specific cytotoxic effect in vitro could be explained by CE emission. It must be noted that, in several studies showing the absence of non-specific cytotoxicity, clonogenic assays were performed using a protocol different from the one followed in the present study. This previous protocol included two consecutive steps. First, cells were exposed to the radionuclide for several hours before centrifugation and washes. Then, cells were seeded at low concentration for clonogenic assay. It is likely that the first step might have led to the loss of the most damaged cells, and therefore, cell survival was only measured for the most viable cells that had not been eliminated by washing and that could adhere to the flask bottom. Moreover, the non-specific irradiation contribution could be different in vivo, particularly considering that, in the present in vitro experiments, cells were incubated with RICs at high activity for 48 h, a situation that might not reflect the in vivo pharmacokinetics of ^111^In-RICs.

Functionalisation with TAT_1-3_ peptides did not significantly modify [^111^In]In-trastuzumab cytotoxicity in A-431_CEA-luc_ cells and slightly increased it in SK-OV-3_1B9_ cells (*p* < 0.01). The observation that the non-specific [^111^In]In-IgG-TAT_1-3_ was as cytotoxic as [^111^In]In-trastuzumab-TAT_1-3_indicates that TAT conjugation is associated with the loss of mAb specificity. The lack of or limited additional cytotoxicity provided by functionalisation with TAT_1-3_ could be explained by the low increase of activity localisation in the nucleus (Ã_nuclear_) after cell exposure to [^111^In]In-trastuzumab-TAT_1-3_. Ã_cellular_ increase in A-431_CEA-luc_ and in SK-OV-3_1B9_ cells exposed to [^111^In]In-trastuzumab-TAT_1-3_ was associated with non-specificity because it also increased in cells incubated with [^111^In]In-IgG- TAT_1-3_. However, the Ã_cellular_ increase in SK-OV-3_1B9_ cells was lower than in A-431_CEA-luc_ cells, suggesting that the higher HER2 expression level of this cell line prevented the passive internalisation of the TAT-RIC.

Functionalisation of radiolabelled mAbs with 17 to 22 NLS/mAb (NLS_17-22_) resulted in a drastic reduction of the surviving fractions for both [^111^In]In-trastuzumab-NLS_17-22_ and [^111^In]In-IgG-NLS_17-22_. Again the addition of NLS peptides was associated with a loss of specificity. Reducing the number of NLS to 5–10 (NLS_5-10_) increased RIC specificity and maintained high cytotoxicity. Nevertheless, even functionalisation with NLS_5-10_ was associated with non-specific cytotoxicity in A-431_CEA-luc_ cells because [^111^In]In-IgG-NLS_5-10_ was more cytotoxic than [^111^In]In-trastuzumab (*p* = 0.001), but the difference between the specific and non-specific [^111^In]In-mAb-NLS_5-10_ effects was more pronounced than for [^111^In]In-mAb-TAT_1-3_. Cell fractionation experiments showed that functionalisation with NLS_5-10_ was accompanied by an increase of the Ã_nuclear_ values for both [^111^In]In-trastuzumab-NLS_5-10_ and [^111^In]In-IgG-NLS_5-10_ in A-431_CEA-luc_ cells and, to a lower extent, in SK-OV-3_1B9_ cells. Nevertheless, the most striking observation was that the functionalisation of [^111^In]In-trastuzumab with NLS_5-10_ significantly increased Ã_cellular_ in both A-431_CEA-luc_ (× 4.7) and in SK-OV-3_1B9_ (× 1.8) cells, with a similar trend also for [^111^In]In-IgG (×52.5 in A-431_CEA-luc_ cells; ×37.2 in SK-OV-3_1B9_ cells), highlighting again the lack of specificity of the NLS-mediated uptake. It has been proven that NLS or TAT peptides pass through plasma membranes in a non-receptor-dependent manner thanks to their highly cationic properties. In the context of our study, NLS or TAT peptides’ addition to mAbs will generate a highly cationic macromolecule that will passively penetrate cells or accumulate onto negatively charged cell-surface proteins. Our results showed an increase of cell uptake when the mAb is conjugated with NLS or TAT peptides suggesting a correlation with the addition of NLS or TAT.

Moreover, the Ã_nuclear_ versus Ã_extranuc_ ratio with [^111^In]In-trastuzumab-NLS_5-10_ was similar to that obtained with [^111^In]In-trastuzumab, suggesting that the NLS peptide did not improve the nuclear translocation of ^111^In-RIC once internalised in the cytoplasm (Figure 4).

These results were supported by the fluorescent detection of immunoconjugate foci, mainly in the cytoplasm, and of very few NLS- and TAT-functionalised immunoconjugate foci in the nucleus. This low nuclear localisation might be due to endosomal–lysosomal entrapment, leading to RIC hydrolysis before they can reach the nucleus and might constitutes a key limiting factor for bringing AEEs into the nucleus. Several groups have developed attractive methods to promote endosomal escape. For instance, in hetero-functional constructs, a target-specific antibody is conjugated to cholic acid for endosomal escape and an NLS peptide is added for nuclear targeting [30,40]. Moreover, modular nanotransporters consist of a target-specific module, a diphtheria toxin translocation domain as endosomolytic module, an NLS peptide for transport to the nucleus, and the *Escherichia coli* haemoglobin-like protein as a carrier module [28,40] [28,41]. These approaches should be further investigated to bypass Auger electron-carrier endosomal entrapment.

At this stage it was not possible to thoroughly investigate the relationship between clonogenic survival and activity uptake in the different cell compartments. Such study would require a comprehensive dosimetric approach (out of the scope of this study) and an assessment of the possible contribution of bystander effects. Moreover, it is important to keep in mind that the cell fractionation experiments were done using activities of 0.1 MBq/mL, and therefore, the only relationship that could be investigated would be with the clonogenic survival data measured at 0.5 MBq/mL.

Interestingly, in a previous study [15], we showed that, in HCT116 cells exposed to [^125^I]I-UdR, Ã_nuclear_ was about 441 Bq.s and Ã_cellular_ was 455 Bq.s. Using the Bliss model and considering that cytotoxicity was only due to radioactivity in the nucleus, we calculated that 80% of cell killing was due to ^125^I located in the DNA [15], while here it was 20% (at 0.5 MBq/mL, a test activity close to the one used for determining the Ã values and for which non-specific irradiation can be neglected) but with about 1.8 (786/441) more decays. This would mean that one decay of ^125^I is 7.2 times more efficient than one decay of ^111^In. If we consider that the energy deposited per decay and due to Auger electrons is about 8.7 keV (^125^I) and 5.4 keV (^111^In) in a 5 µm diameter sphere, then ^125^I deposits about 1.6 times more “high LET” energy than ^111^In. Moreover, a correction factor also needs to be introduced to counterbalance the localisation differences (nuclear but not bound to DNA versus nuclear and bound to DNA). As a relative biological effectiveness value of seven has been proposed when ^125^I is bound to DNA [42,43] and of four [7] when unbound, then the factor that needs to be introduced here is 1.7. Additionally, HCT116 cells are about 1.8 times more radiosensitive than A-431 cells at a dose of 2 Gy [44]. In conclusion, ^111^In incorporated in our RICs should be 8.8 times (2.1 × 1.6 × 1.7 × 1.8) less efficient than [^125^I]I-UdR, a value that can be compared to the value of 7.2 found above.

The preliminary analysis of DNA damage yield (γH2AX foci) in A-431_CEA-luc_ cells exposed to 4 MBq/mL RIC showed that the highest levels of γH2AX foci were observed with [^111^In]In-trastuzumab-NLS_5-10_ and [^111^In]In-IgG-NLS_5-10_, which are associated with the highest cytotoxicity and the highest Ã_nuclear_ and Ã_cellular_ values, although the transient DNA DSB yield cannot be strictly correlated with cell survival. As DNA breaks can be considered to be produced mostly by radiation, we investigated the relationship between the cumulative number of γH2AX foci measured over 144 h and the Ã_nuclear_ and Ã_cellular_ values. However, it must be kept in mind that DNA DSBs can also be produced by radioactivity contained in the culture medium or in neighbouring cells, two parameters not considered here. Moreover, the efficacy of DNA repair occurring at the same time as irradiation [16] could be influenced by the nature of DNA DSBs induced by radioactivity localised in or outside the nucleus. Despite these approximations, we found a relationship between the cumulative number of DSBs and the Ã_nuclear_ and Ã_cellular_ values, although none of these parameters seemed to fit the data best. It must be also noted that some DNA DSBs can be produced by a delayed mechanism involving cell membrane irradiation. We showed previously that Auger electron-mediated irradiation of the cell membrane induces lipid raft formation and the subsequent activation of signalling pathways, leading to the formation of ROS via NF-kB and DNA damage [15,16,17]. We could hypothesise that the high Ã_extranuc_ value of [^111^In]In-trastuzumab-NLS_5-10_ (2412 Bq.s) is related to HER2 binding at the cell membrane, which is not possible with [^111^In]In-IgG. However, this hypothesis needs to be further assessed because we do not know whether enough energy is deposited at the cell membrane.

## 5. Conclusions

In this study, we showed that the functionalisation of trastuzumab with NLS_5-10_ or TAT_1-3_ was accompanied by a rather limited increase in nuclear activity level and therefore, without a drastic decrease in clonogenic survival after exposure to ^111^In-RIC. We obtained similar results with the functionalised non-specific IgG, suggesting that the process is associated with a loss of specificity. This loss of specificity was stronger with TAT_1-3_ than NLS_5-10_. This low nuclear uptake might be due to endosomal–lysosomal entrapment, leading to RIC hydrolysis before they reach the nucleus, and alternative methods to favour endosomal escape are required. Our results also indicate that the non-specific toxicity of ^111^In can be a limiting factor for AEE-based TRT, but this needs to be further assessed in vivo in mice for the reasons mentioned above. The comparison with [^125^I]I-UdR indicates that the amount of energy released in the nucleus by modulating activity uptake or by choosing a better AEE should be multiplied by 10 to obtain significant cell killing (e.g., 99%).

## Figures and Tables

**Figure 1 pharmaceutics-13-00980-f001:**
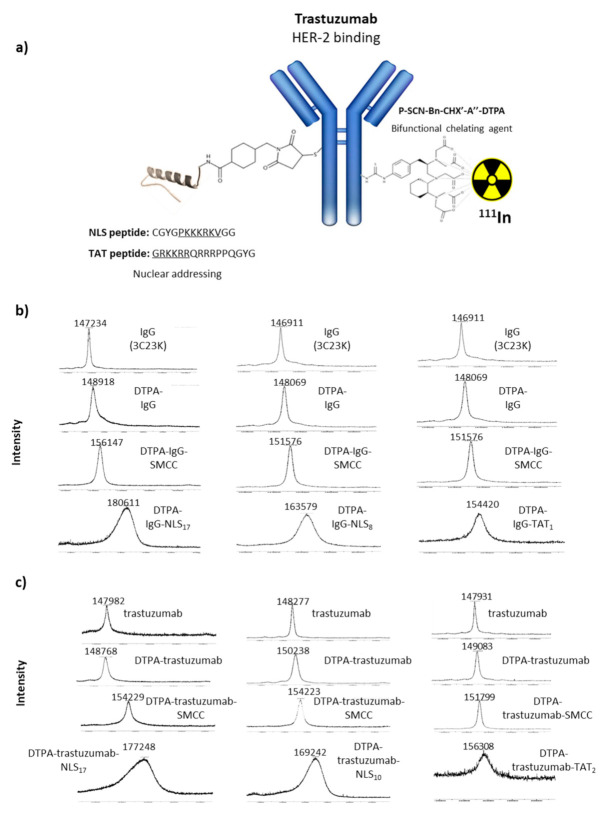
MALDI–TOF analysis of immunoconjugates. (**a**) Schematic representation of the DTPA-mAb-sulphoSMCC-TAT and DTPA-mAb-sulphoSMCC-NLS immunoconjugates. (**b**) MALDI–TOF analysis of IgG, DTPA-IgG, DTPA-IgG-sulpho-SMCC and DTPA-IgG-sulpho-SMCC-TAT, and DTPA-IgG-sulphoSMCC-NLS immunoconjugates. (**c**) MALDI–TOF analysis of trastuzumab, DTPA-trastuzumab, DTPA-trastuzumab-sulpho-SMCC and DTPA-trastuzumab-sulpho-SMCC-TAT, and DTPA-trastuzumab-sulphoSMCC-NLS immunoconjugates.

**Figure 2 pharmaceutics-13-00980-f002:**
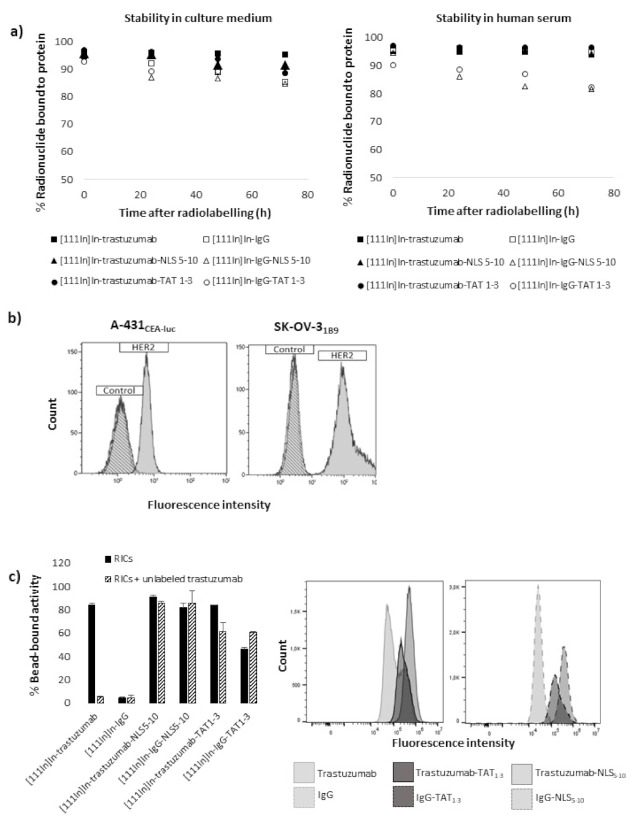
Stability and immunoreactivity of immunoconjugates, HER2 expression. (**a**) In vitro stability of radioimmunoconjugates. (**b**) HER-2 expression on A-431_CEA-luc_ and SK-OV-3_1B9_ cells evaluated by flow cytometry. (**c**) Immunoreactivity assessment using HER2-coated beads (**left** panel). Flow cytometry assessment of cell binding using a secondary FITC-labelled anti-human IgG antibody (**right** panel).

**Figure 3 pharmaceutics-13-00980-f003:**
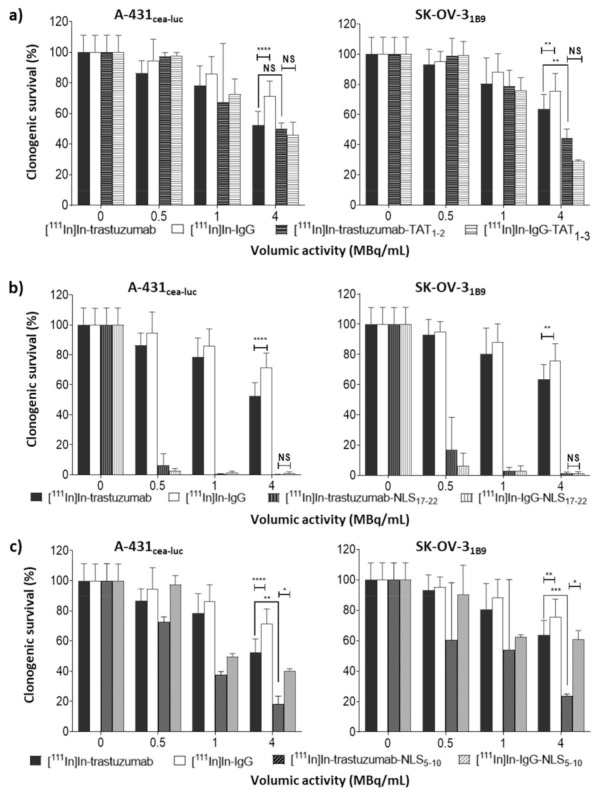
In vitro clonogenic cell death of radioimmunoconjugates. Clonogenic survival was assessed in A-431_CEA-luc_ and SK-OV-3_1B9_ cells 12 days after a 48 h exposure to [^111^In]In-trastuzumab or [^111^In]In-IgG, and (**a**) [^111^In]In-trastuzumab-TAT_1-3_ or [^111^In]In-IgG-TAT_1-3_; (**b**) [^111^In]In-trastuzumab-NLS_17-22_ or [^111^In]In-IgG-NLS_17-22_; (**c**) [^111^In]In-trastuzumab-NLS_5-10,_ or [^111^In]In-IgG-NLS_5-10_ (0–4 MBq/mL; 0–10.8 µg/mL). Data are the mean ± SD. Experiments were performed at least three times in triplicate. * *p* ≤ 0.05, ** *p* ≤ 0.01, *** *p* ≤ 0.001, **** *p* ≤ 0.0001.

**Figure 4 pharmaceutics-13-00980-f004:**
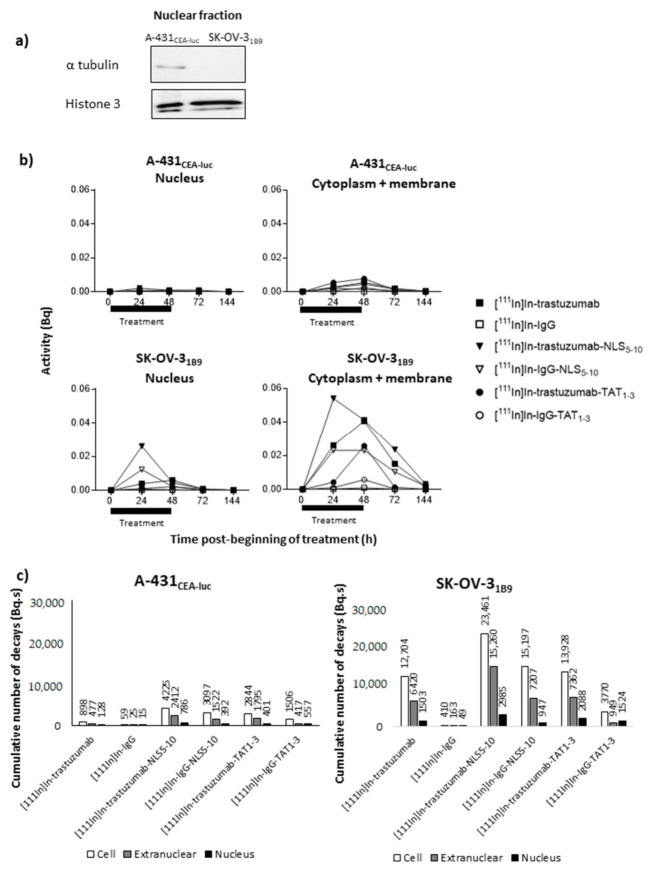
Evaluation of the nuclear and extranuclear uptake of radioactivity (**a**) Expression of histone 3 and α-tubulin was determined in the nuclear fraction by Western blotting to validate the nuclear-fraction isolation technique. (**b**) Cells were exposed to the indicated ^111^In-radioimmunoconjugates (0.1 MBq/mL) for 48 h. At different times post-treatment onset (from 24 h to 144 h), cells were counted and fractionated, and activities were measured in whole cells, in the nuclear fraction and in the extranuclear fraction. Using the AUC, (**c**) the cumulative number of decays per cell or nucleus for the different RICs was calculated in the whole cell (Ã_cellular_), in the cytoplasm (Ã_extranuclear_), and in the nucleus (Ã_nuclear_).

**Figure 5 pharmaceutics-13-00980-f005:**
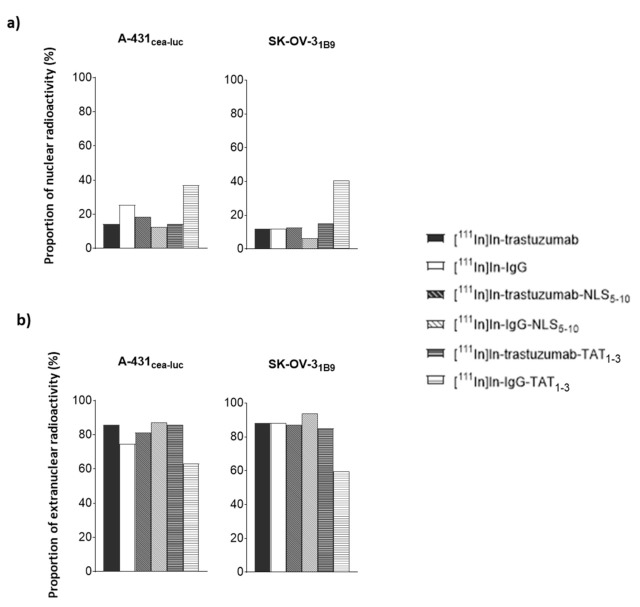
Percentage of: (**a**) nuclear and; (**b**) extranuclear activity. The percentage of activity in the nuclear and extranuclear fraction was calculated for each RIC based on the cumulative number of decays per cell (Ã_nuclear_ and Ã_cellular_).

**Figure 6 pharmaceutics-13-00980-f006:**
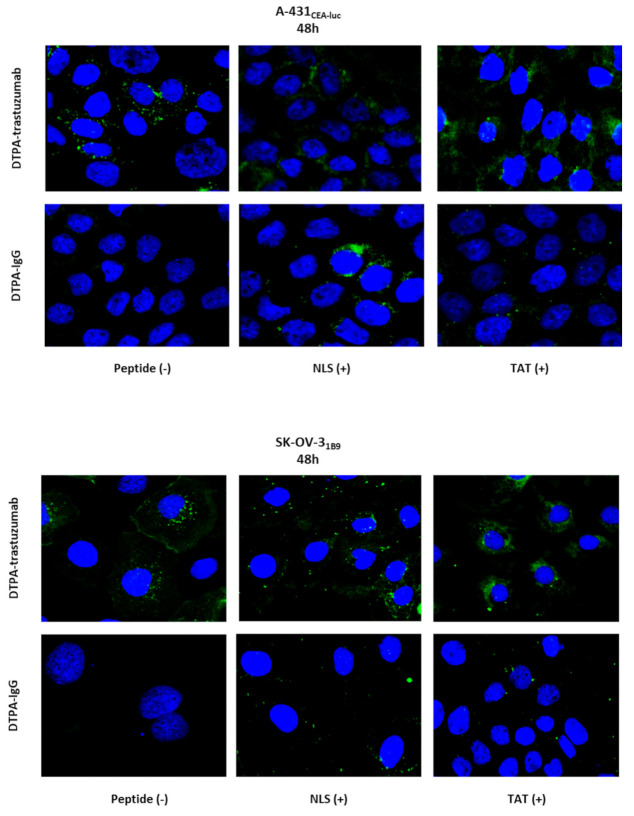
Immunofluorescence analysis of immunoconjugates 48 h after incubation onset. A-431_CEA-luc_ and SK-OV-3_1B9_ cells were incubated with DTPA-trastuzumab, DTPA-trastuzumab-NLS_5-10_, DTPA-trastuzumab-TAT_1-3_, or DTPA-IgG, DTPA-IgG-NLS_5-10_, or DTPA-IgG-TAT_1-3_ (10.8 µg/mL) for 48 h. Cells were fixed, permeabilised and incubated with an AlexaFluor-488-labelled anti-human IgG antibody. Images were acquired using an ×63 or ×40 objective.

**Figure 7 pharmaceutics-13-00980-f007:**
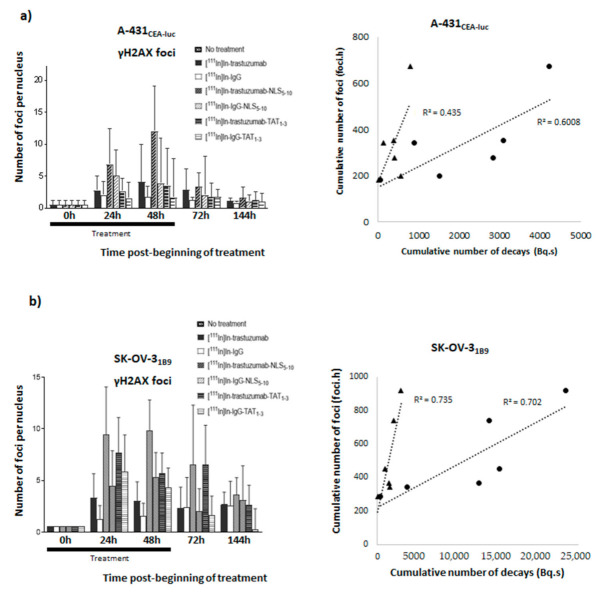
Quantification of γH2AX foci. γH2AX foci formation was evaluated at various time points (24 h, 48 h, 72 h, and 144 h) in (**a**) A-431_CEA-luc_ and (**b**) SK-OV-3_1B9_ cells after incubation with 4 MBq/mL of each RIC. Briefly, cells were fixed, permeabilised, and incubated with an anti-γH2AX antibody followed, after washing, by a goat anti-mouse IgG FITC conjugate. Images were acquired using a ×63 or ×40 objective, and at least 100 nuclei were analysed per immunoconjugate. Data are the mean ± SEM. The cumulative number of γH2AX foci (foci.hour) was expressed as a function of the cumulative number of decays (Bq.s) in the nucleus (triangle markers) and in the whole cell (circle markers).

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
