# Peer review of "Targeted Radionuclide Therapy Using Auger Electron Emitters: The Quest for the Right Vector and the Right Radionuclide"

_pharmaceutics, 2021, doi:10.3390/pharmaceutics13070980_

Round 1

Reviewer 1 Report

The article provided by Dr. Pouget et al., presents Targeted radionuclide therapy using Auger electron emitters. Authors investigated a new type of radiolabeled immunoconjugates, which are designed to target nucleus in tumor cells. It seems that their studies could not provide desired results, however the hypothesis, experimental procedures, and processes are quite interesting. A good research doesn't always lead to good results - These kinds of studies could provide useful information to the researcher. But I think this article should be revised for publication in the journal. Please add some discussion or further results for the following requests.

1) In the introduction, add advantages of AEES compared with alpha- or beta-particles appled TRT. Currently, the comparision and the reasons why the authors investigated AEES rather than alpha- or beta-TRT are short. 

2)  Trastuzumab is consists of more than 1,300 amino acids. While the targeting peptide in this study is much smaller than the antibody. Can the peptides conjugated in the larger biomolecule work properly and contribute to target nucleus? 

3) Conjugation reactions of peptide or chelator (DTPA) have been done in non-specific manner. That process may result in decrease of biological activity of the antibody. Thus, the design of probes should be modified not to (or as low as possible) affect the biological activity.   

4) When Trastuzumab is mixed with the large excess of the bifunctional liker, are there any aggregates or precitation of antibody? 

5) The in vitro stability of radiotracer (i.e. demetallation of In-111 in the biological media) should be added in the main text.  

6) Some of the results could not meet the hypothesis of the research. Please discuss possible alternative strategies or detailed next plans to achieve the final goal.  

Reviewer 2 Report

Review: Targeted radionuclide therapy using Auger electron emitters: looking for the right vector, looking for the right radionuclide

This paper reports the features of 111In labeled trastuzumab radioimmunoconjugates when it comes to cellular localization and effectiveness as potential auger electron therapy radiopharmaceuticals.  The results and methodology makes important contributions to the academic knowledge in this field. The paper is generally well structured and the results clearly described. The conclusions drawn are reasonably established by the results presented.

Below are some minor comments.

Abstract: “111In-labelled” does not need [111In] since it is a concept and not a radiopharmaceutical. See Coenen et. al. “Consensus nomenclature rules for radiopharmaceutical chemistry”.  

Page 2: Pt-193m and Pt-195m are attractive AEEs, however, there are more interesting and more readily available radionuclides that can potentially be used. The authors should reference the article from Filosofov et. al in Nuclear Medicine Biology 2021.

Discussion:

  1. There needs to be an in depth detailed discussion comparing the results of this study to the results in reference 7 (Constantini et. al.). That paper describes a clear higher surviving fraction when 111In-NLS-IgG is used compared to 111In-NLS-Trastuzumab.
  2. It looks to this reviewer like the conjugation with NLS or TAT gives the cell uptake ability more so than Trastuzumab vs “IgG”. The authors repeatedly concludes that this is non-specific uptake, but why would the NLS modification lead to such an increase in non-specific uptake? This should be explained in the discussion.

Conclusion: The section that starts with “Several gtoups have developed attractive methods…. “ is not a conclusion of this paper and can be moved to the discussion.

Reviewer 3 Report

In this work, the authors have studied the effect of modifying trastuzumab and a nonspecific IgG with NLS and TAT peptides to investigate whether such a conjugation will enhance nuclear delivery of AEEs, in this case 111In. They obtained a bunch of negative results (for example, better cell killing with nonspecific conjugate that specific one) and this reviewer is not certain whether this work will be of interest to investigators working this field.  The authors need to address the following issues.

  1. The writing has a considerable room for improvement and is riddled with a lot of mistakes and nonsensical sentences. A version of the manuscript with some annotations is provided; however, the authors need to seek assistance from an expert, who has mastery of English language, especially pertaining to scientific literature.
  2. Page 3, section 2.1: HER2 expression levels in A431 cell line is more than two orders of magnitude lower compared with SKOV3 (PLoS ONE 6(9): e24739. doi:10.1371/journal.pone.0024739). Why was it even considered for this study, which used an anti-HER2 mAb?
  3. Page 6, Section 3.1: Figure 1 subsections need to be rearranged. MALDI-TOF figure should be 1a etc.
  4. The sequence of TAT peptide shown in Figure 1 does not have a cysteine, which is needed for conjugation with maleimide.
  5. MALDI-TOF data is a bit confusing. While it seems some figures are duplicated (for example, top row, left and right one seem identical except for the fact that MW is off by a few Da), data for some are missing (for example, DTPA-trastuzumab-TAT).
  6. Before undertaking clonogenic assays, the specificity of modified trastuzumab to HER2 should have been determined but they did not.
  7. Page 13, first para, second sentence from bottom: Should it be “HER2-specific/non-specific” vs “non-specific/ HER2-specific”?
  8. Page 13, last para: Data for NLS and TAT conjugates are not given in Figure 5.
  9. Page 14, section 3.9: Why was data for Figure 6 generated with only A-431 cells? Not sure the merit of using this cell line at all as it does not overexpress HER2 (see issue 2 above).
  10. Discussion, 1st para: References 30 and 31 do not seem to have anything about HER2, yet the statement is with reference to HER2.
  11. Page 16, 1st para, last three sentences: Do these have any relevance to the discussion above that?
  12. Page 17, 2nd para: Do the authors expect HER2 specificity mediated by NLS?

Round 2

Reviewer 1 Report

The authors revised the manuscript properly and have addressed most of concerns sucessfully. I'd recommend accept of this manusctipt in the journal.   

Author Response

We thank Reviewer 1.

Reviewer 3 Report

The authors have addressed most issues satisfactorily.  Certainly, the manuscript is much more pleasant to read than it was originally.  The authors need to address a few more issues:

  1. Page 5, line 124: resupended or redissolved? Were the peptide not water-soluble?
  2. Page 9, line 254: “bifunctional conjugation” vs “reticulating”?
  3. Page 9, 2nd para: Any reason why the antibody was modified with only 1-3 TAT peptides vs 17-22 NLS peptides although the stoichiometry of peptides and sulfo-SMCC were the same?
  4. Page 9, line 261: “radionuclide or 111In” vs “RIC”.
  5. Page 9, line 264: Shouldn’t it be “predominantly” vs “partly” here as well as a couple of other places?
  6. Figure 1a: The linkage between maleimide and antibody should be thioether, not a disulfide. Also, underline the amino acids in NLS sequence that are involved in nuclear translocation.
  7. Figure 1b and 1c: NLS8 vs NLS10 and TAT1 vs TAT2. Is this intentional or a mistake?
  8. Figure 1 caption: Part c is missing.
  9. Data for NLS17-22 is missing in Figure 2. Was its stability not performed?
  10. Figure numbers are all mixed up. For example it should be Figure 3 vs 2 given in the last para of page 12.  Need to double check all figure numbers in the text.
  11. Provide X-axis title for Figure 3 panels.
  12. Page 23, line 433: “…..this can NLS…” not sensible.

Author Response

Reviewer 3:

The authors have addressed most issues satisfactorily.  Certainly, the manuscript is much more pleasant to read than it was originally.  The authors need to address a few more issues:

We thank Reviewer 3 for his extensive review of our paper that have clearly improved it. Modifications in the text are in green.

  1. Page 5, line 124: resupended or redissolved? Were the peptide not water-soluble?

Response: We thanks the reviewer for highlighting this distinction. Correction was made. Indeed, both NLS and TAT peptides are water soluble.

  1. Page 9, line 254: “bifunctional conjugation” vs “reticulating”?

Response: Sulfo-SMCC is used to cross-link our antibodies to NLS/TAT peptides, we maintain that in this context the term reticulating is correct.

  1. Page 9, 2nd para: Any reason why the antibody was modified with only 1-3 TAT peptides vs 17-22 NLS peptides although the stoichiometry of peptides and sulfo-SMCC were the same?

Response: After numerous bioconjugation assays, we were unable to obtain more than 3 TAT per mAb no matter the excess of TAT added. Compared to bioconjugations with NLS, which number per mAb strongly depends on the molar excess added, we could notice the apparition a slight precipitate over time. Therefore, we hypothesize that this lower functionalisation rate is due to a poorer solubility of the TAT peptide.

  1. Page 9, line 261: “radionuclide or 111In” vs “RIC”.

Response: Correction was made.

  1. Page 9, line 264: Shouldn’t it be “predominantly” vs “partly” here as well as a couple of other places?

Response: The text has been modified accordingly.

  1. Figure 1a: The linkage between maleimide and antibody should be thioether, not a disulfide. Also, underline the amino acids in NLS sequence that are involved in nuclear translocation.

Response: Figure 1 has been modified to highlight the thioether linkage. NLS amino acids sequence involved in nuclear translocation is now underlined in Figure 1.

  1. Figure 1b and 1c: NLS8 vs NLS10 and TAT1 vs TAT2. Is this intentional or a mistake?

Response: Typically, we obtained from 1 to 3 TAT/mAb or 5 to 10 NLS/mAb for similar bioconjugation conditions. Therefore, these values are not a mistake, they translate the actual peptide/mAb ratios for the bioconjugations whose results are presented in these figures.

  1. Figure 1 caption: Part c is missing.

Response: Part c has been added in the legend.

  1. Data for NLS17-22 is missing in Figure 2. Was its stability not performed?

Response: The stability of 111In-labeled mAbs-NLS17-22 has not been performed. We do not expect any significant difference in stability with these radiopharmaceuticals as the addition of NLS and TAT peptides (in lower proportion) did not negatively impact the stability of our RIC.

  1. Figure numbers are all mixed up. For example it should be Figure 3 vs 2 given in the last para of page 12.  Need to double check all figure numbers in the text.

Response: All the figures numbers has been checked and modified if necessary.

  1. Provide X-axis title for Figure 3 panels.

Response: All the X-axis title have been added for more clarity.

  1. Page 23, line 433: “…..this can NLS…” not sensible.

Response: Correction was made. “This can” has been removed from the text. Now, the sentence is “NLS or TAT peptides addition to mAbs will generate a highly cationic macromolecule that will passively penetrate cells …..”

Round 3

Reviewer 3 Report

Issue 5 of previous review: Need to change "partly" to "predominantly" at two more places

Although they stated the issue 6 has been addressed, it is not.  It is still a disulfide bond and the NLS amino acids are not underlined.

Author Response

Reviewer 3:

Comments and Suggestions for Authors

  1. Issue 5 of previous review: Need to change "partly" to "predominantly" at two more places

The text was modified accordingly (highlighted in blue, lines 270 and 398)

  1. Although they stated the issue 6 has been addressed, it is not.  It is still a disulfide bond and the NLS amino acids are not underlined.

We are sorry the figure was not updated in the text. The modification is done.
